# The Prognostic Importance of ctDNA in Rectal Cancer: A Critical Reappraisal

**DOI:** 10.3390/cancers14092252

**Published:** 2022-04-30

**Authors:** Edina Dizdarevic, Torben Frøstrup Hansen, Anders Jakobsen

**Affiliations:** 1Danish Colorectal Cancer Center South, Department of Oncology, University Hospital of Southern Denmark, 7100 Vejle, Denmark; torben.hansen@rsyd.dk (T.F.H.); anders.jakobsen@rsyd.dk (A.J.); 2Department of Regional Health Research, University of Southern Denmark, 5000 Odense, Denmark

**Keywords:** rectal cancer, colorectal cancer, ctDNA, liquid biopsy, biomarkers, QUIPS

## Abstract

**Simple Summary:**

An individualized treatment approach is necessary to improve survival and quality of life in rectal cancer. Tools to stratify patients are missing, but ctDNA seems to be a good candidate. Current results are sparse, conflicting and characterized by lack of a uniform approach. As the interpretation of results is dependent on the quality of reporting, we aimed to address this issue in our paper. Our results indicate an association between ctDNA status and outcome. In six out of nine papers, bias was low. However, a conclusion is dubious because of the heterogeneity among the studies and lack of standardized methods. Studies addressing these issues are warranted.

**Abstract:**

The treatment of locally advanced rectal cancer (LARC) has evolved during the last decades, but recurrence remains a problem. Circulating tumor DNA (ctDNA) may result in an individualized treatment approach with improved survival and quality of life, but diverging results impede further development. In this systematic review, we addressed the quality of reporting and its impact on the interpretation of ctDNA results. We performed a systematic literature search using subject headings and search terms related to ctDNA and rectal cancer. The Quality of Prognostic Studies (QUIPS) tool was used to assess bias. Nine studies, with substantial heterogeneity, were included in the analysis. Three out of nine articles had moderate or high risk of bias. No association was found between treatment response and ctDNA status at baseline. There was a negative association between ctDNA positivity at baseline, before and after surgery and survival. The ctDNA status may be of importance to the long-term prognosis, but the area of research is new and is short of dedicated studies. There is an obvious need for standardization in ctDNA research, and the issue should be addressed in future research.

## 1. Introduction

The treatment of locally advanced rectal cancer (LARC) has evolved from surgery alone to experimental trimodality treatment comprising neoadjuvant chemoradiotherapy (nCRT) followed by surgery. Modifications of radiation doses and treatment schedules have been presented with different rates of pathological response but without improvement of long-term survival [1,2,3,4,5]. Complete pathological response remains an important prognostic factor but is a poor surrogate endpoint for overall survival [6].

The achievement of complete pathological response in surgical specimens suggested the possibility of organ preservation. This approach has been offered to a selected group of rectal cancer patients with clinical complete response (cCR) after chemoradiation [7]. Several studies have reported excellent oncological outcome and high quality of life [8,9,10,11], but a relatively high fraction will recur during follow-up. It remains unclear how patients should be selected for surgery versus observation after chemoradiation. Total neoadjuvant therapy is a new approach in rectal cancer, but the concept needs further validation before it comes to clinical utility [12,13].

The treatment of rectal cancer calls for new markers, which alone or in combination can guide a more individualized treatment approach. 

Circulating tumor DNA (ctDNA) is a new biomarker that has gained rapidly increasing interest. In many aspects, it seems to fulfill the criteria of an ideal tumor marker. It is easily accessible for repeated measurement with little discomfort to the patient, reflects important biological characteristics and seems to overcome the issue of tumor heterogeneity [14,15]. There is a steep increase in the number of publications, witnessing a considerable clinical interest. In colon cancer, several prospective trials are investigating if ctDNA can identify patients with non-metastatic colon cancer, with high risk of recurrence, that may benefit from adjuvant chemotherapy [16,17,18,19]. However, in rectal cancer, the influence of ctDNA on treatment decisions is still negligible. The diverging results and lack of standardization pose a major problem and impede further development. The situation calls for a comprehensive evaluation of the current literature.

This review aimed to analyze the quality of published studies, especially the risk of bias as evaluated by the international Quality of Prognostic Studies (QUIPS) tool [20].

## 2. Materials and Methods

Our findings are reported according to the PRISMA (Preferred Reporting Items for Systematic Review and Meta-Analysis) guidelines [21].

Studies meeting the following criteria were included: original articles in English with at least 25 participants aiming to investigate the prognostic role of ctDNA in advanced, non-metastatic rectal cancer. Reviews and studies publishing pooled data from rectal and colon cancer were excluded. Quality assessment in the primary selection was not required. 

The search strategy was wide to ensure the inclusion of all relevant publications. The reference lists of included studies or relevant reviews were screened for additional references.

A literature search strategy was established together with a research librarian at the University of Southern Denmark using a mixture of subject headings and terms related to rectal cancer and ctDNA (Table 1). One researcher (ED) performed the literature search in Pubmed, Embase and Scopus with the last search on 14 June 2021.

Search results were uploaded to covidence.org. The first author screened the title and abstract yielded by the search against the inclusion criteria. Full-text review was performed independently by ED and AJ. Reasons for exclusion were recorded, and potential disagreement was solved through discussion. None of the authors were blinded in the process. 

A standardized form was created and tested by ED prior to data extraction. The form was discussed with AJ, and adjustments were made before the final data extraction. To ensure consistent methodology, data extraction was performed by ED and revised by AJ. For each study included, we extracted study characteristics (author, year, sample size, age and sex distribution, stage of disease), ctDNA measurements (times of ctDNA measurement, laboratory methods), follow-up period, and main findings (treatment response, survival outcomes)

### Quality Appraisal

Studies evaluating new potential prognostic markers are recommended to use the Reporting Recommendations for Tumor Markers Prognostic studies (REMARK) criteria to ensure quality and transparency of reporting. The REMARK checklist was applied in the assessment of the studies.

Bias was evaluated with the Quality in Prognosis Studies tool (QUIPS). It evaluates six domains, but it does not provide a scoring system [20]. Instead, each domain is rated as having a high, moderate or low risk of bias. Each of the six domains involves several questions, and the risk of bias can be evaluated based on the answers. The six domains address study participation, study attrition, prognostic factor measurement, outcome measurement, study confounding, and statistical analysis and reporting. The reviewer evaluates whether all six domains should be addressed. The first author and AJ individually assessed the quality of the included studies using QUIPS. Disagreements on study quality were solved through discussion.

Detailed evaluation of the extracted data showed that a valid quantitative synthesis was not possible mainly because of the heterogeneity of the included studies. Therefore, we report the outcomes of interest narratively.

## 3. Results

A total of 552 articles were identified through the search strategy of which 222 were doublets and removed. Thirty-four studies remained after the screening of title and abstract, and 25 studies were excluded at retrieval of full text articles. Reasons for exclusion are described in Figure 1. Only nine studies were eligible for inclusion. We did not find additional papers going through the reference list of included studies.

### 3.1. Quality Appraisal

Six of nine studies had a low risk of bias as assessed using QUIPS. Two of the studies had a high risk of bias in the categories “study attrition” and “study confounding”. One of them also had a high risk of bias in the domain of statistical analysis. Lastly, one of the nine studies had a high risk of bias in all six domains. The results of the quality assessment are presented in Figure 2. 

Only one study reported the use of the REMARK checklist. 

### 3.2. Study Characteristics

Important characteristics of the nine included articles are described in Table 2. Studies were published from year 2017 and forward. The follow-up period varied on a large scale, and some did not report the timeframe. The research design of the studies was retrospective [22,23,24,25,26] or prospective [27,28,29,30]. The number of included patients differed between studies ranging from 32 to 159 patients. In total, the studies report data on 778 patients, predominantly male, with a median age of 58 to 64 years. There was a preponderance of stage III disease (ranging from 68 to 100%). One study described only the clinical T-stage, pathological T and N-stage [26]. 

Treatment regimens were heterogeneous across the studies. However, long course radiotherapy (RT) given prior to surgery was relatively consistent. In the study by Murahashi et al., some patients received short course (25 gy/5 fr), while others received long course RT [30]. In the study by Appelt et al., a brachytherapy boost of 2 × 5 Gy was delivered in a fraction of the patients. The chemotherapy treatment consisted of concomitant fluoropyrimidine in most studies. Treatment with adjuvant chemotherapy was according to institutional guidelines. Chemotherapy, prior to the neoadjuvant treatment, was given in few of the studies [23,24,30]. Hence, there was a large variation in the treatment strategies. Organ preservation was an option in two of the included studies, and a small subgroup of patients was treated accordingly [25,30].

### 3.3. Measurement of ctDNA

The methods for ctDNA detection differed considerably. Three studies applied droplet digital PCR (ddPCR); two [23,25] targeted mutations and one study explored methylated Neuropeptide Y [22]. One trial applied conventional PCR [27]. Pazdirek et al. performed denaturing capillary electrophoresis (DCE) for mutation detection and crosschecked negative samples with a beaming assay directed at KRAS-specific ctDNA. Four studies applied next-generation sequencing (NGS) with different targets and sequencing depth.

Four [23,25,27,28] out of nine trials analyzed the tumor tissue, and two of them used a tumor-informed approach. 

The three studies that applied ddPCR for ctDNA detection reported their results as fractional abundance and ctDNA as a mutated fraction of the total amount of circulating DNA [22,23,25]. Samples were considered positive with >2 droplets present [22,25]. Sclafani et al. set a sensitivity cut-off for ctDNA detection at the lower limit of 0.02%.

Murahashi et al. report their findings as mutant allele fraction (MAF) with a cut-off value of 0.15%. The MAFs represent the number of mutant reads divided by the total number of reads at a specific genomic position. Tie et al. used a similar approach with a cut-off value of 0.1%, but a comparison of mutation frequencies was accomplished with a permutation test, and results were categorized as either positive or negative. Zhou et al. applied the same cut-off level.

Ji et al. expressed their findings as tumor mutation burden in blood (bTMB) based on variant allele frequencies (VAF).

### 3.4. Outcome

Seven out of the nine included studies reported treatment response; none of them found a significant difference between ctDNA status at baseline and treatment response. Four studies reported similar findings between ctDNA status after surgery and treatment response. However, Khakoo et al. report that poor responders were more likely to have detectable ctDNA after CRT. These results were supported by the findings of Tie et al. with detectable post-surgery ctDNA and tumor and node stage (ypT3-4 and ypN1-2).

The median follow-up time ranged 18.8 months to 10.6 years. Four studies did not describe the length of the follow-up period [23,26,28,30].

The reporting on survival outcomes varied largely between the studies. Survival was reported as overall survival (OS), disease-free survival (DFS), progression-free survival (PFS), metastases-free survival (MFS), recurrence-free survival (RFS), and one reports local recurrence-free survival (LRFS).

#### 3.4.1. ctDNA at Baseline

All of the included studies evaluated the association between baseline ctDNA and survival. In five out of nine studies, there was no association between survival and ctDNA status at baseline. Pazdirek et al. found that positive ctDNA status prior to nCRT was associated with a significantly shorter DFS and OS. Appelt et al. results support these findings and report worse 5-year OS in patients with positive ctDNA at baseline. Metastatic disease seemed to be the driver of this difference. The prognostic importance of ctDNA remained true in the regression analysis with a hazard ratio (HR) of 2.08 95% CI (1.23–1.51). Zhou et al. reported significantly worse MFS for patients with positive ctDNA at baseline (*p* = 0.03). Median VAF of mutations (>1%) in baseline ctDNA showed a strong predictive value of distant metastasis (HR 6.5 95% CI (1.67; 25.73)).

Surprisingly, Ji et al. found that patients with low levels of bTMB at baseline had an increased risk of regrowth.

#### 3.4.2. ctDNA after CRT

Six studies evaluated the prognostic impact of ctDNA after CRT and before surgery. Murahashi et al. did not find a significant correlation between ctDNA status and survival, which was similar to the results from Ji et al. Four studies reported a significant difference in the survival outcome according to ctDNA status. Tie et al. reported an RFS at 3 years of 50% and 85% for the ctDNA positive and negative group, respectively. Vidal et al. reported similar results with a 3-year DFS of 66% and 84% for the ctDNA positive and negative group. These results were supported by the findings form the two remaining studies who reported shorter MFS for patients with detectable ctDNA after the completion of CRT [25,29]. 

#### 3.4.3. ctDNA after Surgery

For the post-surgery ctDNA measurements, the results were unanimous. All five studies reported an association between ctDNA positivity after surgery and worse survival. Khakoo et al. reported even worse MFS for patients with ctDNA persistence throughout the treatment. Zhou et al. found a shorter MFS for patients with detectable ctDNA compared to those with undetectable ctDNA. This correlation was found at all time points (during CRT, before and after surgery), and the HR increase at each time point with HR 6.635 95% CI (1.24; 35.50) during CRT, HR 19.82 95% CI (2.03; 193.7) before surgery and HR 25.3 95% CI (1.48; 434.0) after surgery. 

## 4. Discussion

The current literature points toward ctDNA as a promising tumor marker in clinical oncology. In rectal cancer, ctDNA may result in an individualized treatment approach. Reviews published during the recent years have tried to outline the role of ctDNA in rectal cancer, but none of them have addressed the quality of the included studies [31,32]. To our knowledge, our review is the first to address this aspect.

Bias was assessed with the QUIPS tool. Six studies were classified as having low bias, two with moderate risk and one with high risk of bias. The developers of the QUIPS tool suggest that studies with high risk of bias should be omitted from further analysis. Based on this statement, only six out of nine studies in our review would be eligible for the evaluation of ctDNA. Since the population in our review is small and our results are reported narratively, we present all results from the analysis. However, this serves as a clear indicator of the necessity of a quality assessment for future reviews.

There is no evaluation of bias in papers reviewing ctDNA in rectal cancer. The lack of quality appraisal is a problem in many reviews. The literature search of Hayden et al. revealed a rising number of published reviews and a consistent problem with the quality assessment. Almost 50% of the articles generated from their search were excluded because of lacking quality assessment. For the remaining articles, the evaluation varied and was often inadequate. Because of this gap, they developed QUIPS to assess validity and bias in studies investigating prognostic factors [20]. This is the only tool specifically developed for quality evaluation of prognostic studies. Although the tool indicates potential problems with only three out of nine studies, it should be used cautiously, as it does not address all problems, i.e., comparison of methodological issues.

Riley et al. suggest a guide that should be applied when conducting systematic reviews and meta-analyses of prognostic studies [33]. In the future, this systematic approach will lead to fewer reviews with comprehensive analysis of the included literature. As a result, the role of different biomarkers may be clarified faster, and unnecessary research can be omitted.

The REMARK checklist was developed to ensure consistent reporting [34,35]. In spite of the introduction of the checklist, results have been discouraging with no improvement in reporting [36]. This is supported by our findings with only one study reporting the use of the checklist. Hence, there is a risk of continuous research in markers without real progress as to clinical utility.

Another issue is the lack of standardized methods for ctDNA detection and reporting, as this impairs comparison and generalization of results. This is indeed what we see in this review. Characteristics and features of the tumor, which are not fully understood, can aggravate this problem along with other problems as illustrated in the literature [22,23,25,37,38,39,40]. The discussion of the methodology is outside the scope of this review, but a comprehensive review has addressed the current challenges and suggests possible solutions which should be considered in future research [41].

The predictive value of ctDNA varied among the included studies. Detectable ctDNA at baseline does not seem to be a predictor of treatment response. Only two of seven studies report the presence of ctDNA after CRT and a poor response as evaluated by MR or the surgical specimen, respectively [25,27]. These results indicate that ctDNA currently cannot be used to select treatment, i.e., organ preservation or surgery. In general, the results should be carefully considered, since none of the studies report power calculations. This is likely a result of the retrospective study design. For the prospective studies, the ctDNA analyses are exploratory, and it may be rather difficult to make proper assumptions regarding the power calculations. Post hoc power calculations are generally not recommended [42]. Hence, the literature is short of dedicated prospective trials exploring the clinical aspects of ctDNA in rectal cancer. However, in colon cancer, several prospective trials are investigating if ctDNA can predict recurrence and if ctDNA-guided treatment can improve survival [17,18,41,43]. Circulating tumor DNA seems promising in other types of cancers, but results are limited by the relatively small size of the trials [44,45,46].

For the survival outcomes, all of the studies found a correlation between survival outcome and ctDNA at a certain time point. For the baseline measurements, three out of nine reported an association between detectable ctDNA and worse survival. For the pre- and post-surgery analyses, four out of six and five out of five, respectively, found an association between detectable ctDNA and worse outcome. The persistence of ctDNA throughout the course of treatment seems to be a poor prognostic marker. The reported increasing HR at each time point supports this hypothesis. Especially detectable postoperative ctDNA seems to correlate with relapse. Similar results are reported in the treatment of colon cancer and malignant melanoma [44,47].

At this point, it is difficult to conclude as to the prognostic importance of ctDNA in rectal cancer as most studies are small, the statistical assumptions are dubious and the follow-up period is rather short in some of the included studies. There is a substantial risk of both missing a correlation and finding one that would not be present if a longer follow-up period were available. In a review assessing trial extensions using routinely collected data, the authors explored if there were any beneficial or harmful events in the period beyond the original trial time period. They showed that in nearly 50% of the trials, a benefit was observed with extended follow-up, and in nearly 30% of these, the benefits were only significant for this period. Loss of significant benefits was only seen in 7% of the studies [48]. These results highlight the importance of a sufficiently long follow-up period in terms of evaluating interventions, new diagnostic or prognostic tools. For rectal cancer, most local regrowths appears within the first two years, while metastatic disease may appear several years later. Therefore, both the follow-up period and the evaluation of the endpoint should be selected carefully prior to trial initiation [49]. Studies with large cohorts and long follow-up are time consuming and expensive, but they are necessary in terms of clarifying the real value of biomarkers. A surrogate endpoint would be ideal to select optimal treatment and reduce costs and follow-up in rectal cancer. The study published by Jakobsen et al. indicates that ctDNA response may serve as a surrogate marker for OS in several cancers [50]. Organ preservation in rectal cancer as standard treatment could be a possibility if ctDNA response proved to be a reliable surrogate marker for oncological outcome.

This review has several limitations. We included only studies published in English limited to a certain time period. Another limitation is the application of QUIPS, which has not been validated. Lastly, we were not able to perform statistics to support our findings. Additional limitations are related to the included studies i.e., small study populations, variable follow-up period, heterogeneous treatments, methods for ctDNA detection and selected outcomes. However, we do believe that we were able to highlight some of the current issues which should be addressed in future trials.

## 5. Conclusions

Circulating tumor DNA may add additional prognostic and predictive information in rectal cancer. Quality appraisal should be a standard requirement in reviews to ensure consistent data. Currently, we are missing dedicated, preferably randomized, trials exploring ctDNA in rectal cancer. The problems raised in this review call for attention in future research.

## Figures and Tables

**Figure 1 cancers-14-02252-f001:**
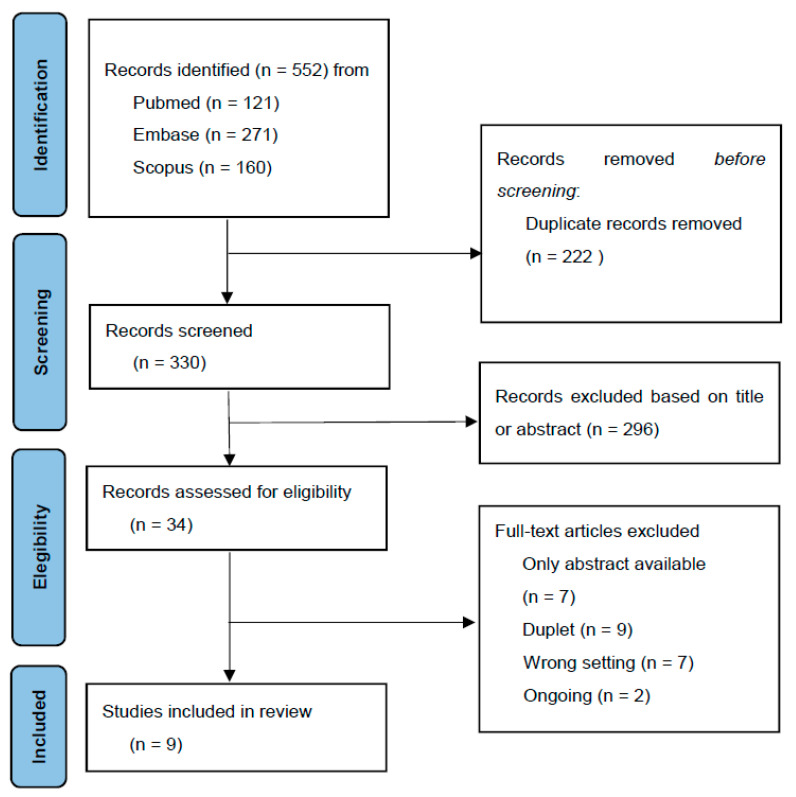
PRISMA flow diagram.

**Figure 2 cancers-14-02252-f002:**
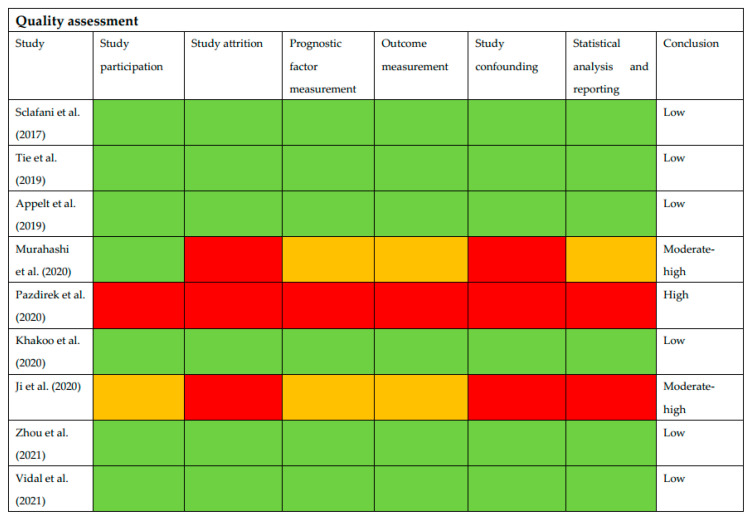
Green indicates low risk, yellow indicates moderate and red indicates high risk of bias according to QUIPS.

**Table 1 cancers-14-02252-t001:** Search strategy.

Database	Search Term 1	Search Term 2
Pubmed	rectal neoplasms (Mesh)	dna methylation (MeSH Terms)
	rectal cancer (Title/Abstract)	“methylated dna” (Title/Abstract) ctDNA (Title/Abstract)
		“circulating tumour dna”(Title/Abstract)
		“circulating tumor dna” (Title/Abstract) cfdna (Title/Abstract)
		“cell free dna” (Title/Abstract)
		“liquid biopsy”
Embase	rectal cancer (keyword)	ctdna (keyword)
	rectum cancer (subject heading)	circulating tumo?r dna (keyword)
	rectal malignancy (keyword)	circulating tumor DNA (subject heading)
		“cell free dna” (keyword)
		cfdna (keyword)
		“methylated dna” (keyword)
		DNA methylation (subject heading)
		liquid biopsy (subject heading)
		liquid biops* (keyword)
Scopus	search within title, abstract	“circulating tumour dna”
	or keyword	ctdna
	rectal cancer	“cell free dna”
	rectal neoplasms	cfdna
		“dna methylation”
		“methylated dna”
		“liquid biopsy

In the left column, all of the items in “search term 1” are related to rectal cancer and are combined with “or”. The “search terms 2” words and phrases are related to ctDNA and are combined with “or”. Search terms 1 and 2 are combined with “and” in each of the databases.

**Table 2 cancers-14-02252-t002:** Study characteristics and main findings.

Study	No.	Patient Characteristics	ctDNA Measurements	Assay	Median Follow-Up	Main Findings
Appelt et al. (2019)	146	age 64 64% male 88% stage III	Baseline	ddPCR	10.6 years (range 9.2–11.5) for OS	There was no association between meth-ctDNA status at baseline and complete (OR 0.64 95% CI 0.11–2.49, *p* = 0.76) or major tumor regression (OR 1.13 95% CI 0.4–3.54, *p* = 1.0). Patients with meth ctDNA had worse OS at 5 years (47% vs. 69% p = 0.02). Regression analysis demonstrated that meth-ctDNA is a strong independent prognostic factor for OS (HR 2.08 95% CI 1.23–1.51) and freedom from distant metastases (HR 2.20 95% CI 1.19–4.07).
Sclafani et al. (2017)	97	age 62.1 60% male 69% stage III	Baseline	ddPCR	NA	There was no difference in CR rate (15.4% vs. 10% OR 1.63 95% CI 0.3–8.96, *p* = 0.57), PFS (HR 0.70 95% CI 0.29–1.7, *p* = 0.43) or OS (HR 0.78 95% CI 0.31–1.96, *p* = 0.60) between ctDNA-positive and ctDNA-negative patients.
Vidal et al. (2021)	62	age 62 64.5% male 100% stage III	Baseline and before surgery	NGS	38 months (range 2.3–51.5) for DFS and OS	No association between ctDNA at baseline and ypCR (*p* = 0.134). Detectable ctDNA at baseline was not associated with DFS (*p* = 0.59) or OS (*p* = 0.38). Patients with detectable ctDNA prior to surgery had increased risk of recurrence (HR 4.0 95% CI 1.0–16.2, *p* = 0.033) and a markedly reduced survival (HR 23 95% CI 2.4–212, *p* < 0.0001). DFS at 3 years was 66% and 84% for ctDNA positive vs. ctDNA negative.
Khakoo et al. (2020)	47	age 59 61.7% male 89% stage III	Baseline, during CRT, before and after surgery	ddPCR	26.4 months (range 19.7–31.3) for MFS	There was no significant difference in response as evaluated by RECIST in patients with detectable ctDNA vs. undetectable at any time point. Poor responders were more likely to have detectable ctDNA after completion of CRT (*p* = 0.03). There was no association between baseline ctDNA status and MFS. MFS was shorter in patients with detectable ctDNA after CRT compared to patients with undetectable ctDNA (HR 7.1 95% CI 2.4–21.5, *p* < 0.001). Persistent ctDNA throughout treatment was associated with worse MFS (HR 11.5 95% CI 3.3–40.4, *p* < 0.001).
Ji et al. (2020)	46	age 58 63% male NA	Baseline, before and after surgery	NGS	NA	No evaluation of an association between ctDNA status and treatment response. There was no significant correlation between ctDNA status at baseline, after CRT or after surgery and tumor recurrence or OS. Patients with low levels of bTMB at baseline had an increased risk of recurrence (*p* = 0.036) while patients with high levels after surgery had a high risk of recurrence (*p* = 0.026).
Tie et al. (2019)	159	age 62 67% male 78% stage II	Baseline, before and after surgery	PCR	24 months (range 1–55) for RFS	ctDNA levels at baseline and after CRT were not associated with pCR. High levels of postoperative ctDNA were associated with ypT3-4 and ypN1-2. There was no difference in RFS between ctDNA positive and ctDNA negative (HR 1.1 95% CI 0.42–3.0) at baseline. Three-year RFS was 50% and 85% for the ctDNA positive and ctDNA negative, respectively (after CRT), and 33% and 87% in the postoperative ctDNA positive and ctDNA negative groups. In regression analysis, ctDNA status remained the strongest independent predictor of RFS (HR 6.0 95% CI 2.2–16, *p* < 0.001).
Pazdirek et al. (2020)	33	age 64 75% male 78% stage III	Baseline and during CRT	DCE	NA	No evaluation of an association between ctDNA status and treatment response. The overall probability of three-year survival was 91.2% in the ctDNA-negative and 71.4% in ctDNA-positive group (7/33 were positive for ctDNA). Positivity for ctDNA at baseline was associated with a significantly shorter DFS (*p* = 0.015) and OS (*p* = 0.010).
Zhou et al. (2021)	104	age 60 64.4% male 77% stage III	Baseline, during CRT, before and after surgery	NGS	18.8 months (range 3.1–21.3) for MFS	There was no association between ctDNA at baseline or during CRT and parameters that reflect tumor response (*p* > 0.05). Positive ctDNA at baseline, during and after CRT, and after surgery was associated with shorter period of time to distant metastasis. HR increased over each time point. Regression analyses showed that only median VAF of mutations at baseline remained an independent predictor of MFS compared to other pretreatment variables (HR 1.27 *p* < 0.001).
Murahashi et al. (2020)	85	age 60 76.5% male 68% stage III	Baseline, before and after surgery	NGS	NA	There was no association between ctDNA status at baseline, after CRT, or after surgery and treatment response (pCR or cCR in patients treated with organ preservation). However change in MAF was associated with response to treatment (OR 7.4 95% CI 1.2–144, *p* = 0.0276). Increased RFS was observed in patients with ctDNA <0.5% vs. >0.5% (HR 17.1 95% CI 1–282, *p* < 0.000).

Abbreviations: ddPCR, digital droplet polymerase chain reaction; ctDNA, circulating tumor DNA; CR, complete response; PFS, progression-free survival; OS, overall survival; RFS, recurrence-free survival; CRT, chemoradiotherapy; pCR, pathological compete response; meth-ctDNA, hypermethylated circulating tumor-specific DNA; NGS, next-generation sequencing; cCR, clinical complete response; MAF, mutant allele fraction; DCE, denaturing capillary electrophoresis; DFS, disease-free survival; MFS, metastases-free survival; bTMB, tumor mutation burden in blood; VAF, variant allele frequencies.

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
