# Peer review of "The Prognostic Importance of ctDNA in Rectal Cancer: A Critical Reappraisal"

_cancers, 2022, doi:10.3390/cancers14092252_

Round 1
Reviewer 1 Report
Thank you for your valuable work, yet there are some flaws to be addressed:
- The manuscript should be revised by a native English speaker
- A section of limitations within the manuscript is missing. I suggest the authors to include it.
- A critical appraisal on clinical impact of ctDNA in clinical current practice is missing. Please read this paper PMID: 31534493
Reviewer 2 Report
Brief summary: The authors present a review of current literature to asses the prognostic utility of ctDNA as a biomarker in locally advanced rectal cancer (LARC) and the quality of reporting. General comments:
- Study/publication inclusion and exclusion criteria are well described, however, the overall objective of the review is not as well articulated (see comments below regarding Abstract and Introduction sections).
- The Abstract states that the review is intended to assess the quality of reporting and its "...impact on the ctDNA results." (line 21-22). It is not entirely clear what is meant by this. The authors report the risk of bias as determined by QUIPS, but do not discuss how such risk of bias should impact the evaluation of the data presented from each study and conclusions drawn from the studies in aggregate. Should results from one study be weighted heavier if there was lower risk of bias and vice versa?
- The authors need to be more clear in the final paragraph of the Introduction (lines 50-59) what the intent of this review is; clearly stating for which disease and in what treatment setting. Lines 54-57 recognize an increasing number of publications and clinical interest in ctDNA as a biomarker and that a comprehensive review of literature is required without distinguishing between those diseases and settings where ctDNA has been more clinically useful and those where it has been less useful, either because of data availability and general interest or unique application. Where does LARC treatment and ctDNA fit into this current landscape and why?
- Discussion does not distinguish cancer types and treatment settings where clinical utility of ctDNA has been more extensively studied and elucidated and how that might inform the value of ctDNA as a biomarker in the treatment/prognostication of LARC. It is often unclear when the authors are discussing ctDNA as a biomarker if it is in the context of LARC clinical prognostication specifically, or in other cancers and their treatment. Since our knowledge around ctDNA - and, hence, the clinical implementation of it as a biomarker - varies significantly among cancer types and treatment scenarios, it is important to state in which disease contexts it has been found to be useful and those it hasn't. For example, line 327-328 states that the "...literature is missing dedicated prospective trials exploring the clinical aspects of ctDNA." This may be true for LARC and many other cancers, but it is not true for all cancers. Authors should use more specific language here. Another example of this ambiguity is in line 338, which states "...it is difficult to conclude if ctDNA is a prognostic marker since the population is small, [etc.]..." - do the authors mean the population, etc. in the studies included in this review, in LARC more generally, or in all cancer types?
- Although the authors acknowledge the challenges in comparing ctDNA results across studies with heterogeneous assay types and detection limits, they do not elaborate on how this informs there approach. Typically, heterogeneity in ctDNA assay type and sensitivity across multiple studies requires an analysis that reduces each study's results to qualitative or categorical summaries (ctDNA detected or not-detected at a given clinically significant time point and ctDNA dynamics reported as increases or decreases over time). The authors do not state how they are able to make cross-study conclusions.
- Line 65 - (section titled "Method") - please describe rationale for using minimum of 25 patients as a cutoff for inclusion in this review. Provide statistical approach/method if used.
- Specific comments for section titled "Measurement of ctDNA"
- Line 206-207 states that four studies used NGS with varying targets and depth. Please specify those studies with citations and briefly describe the NGS approach (custom targets, amplicon vs hybrid-capture, etc) and range of sequencing depths and implications for sensitivity.
- The authors state which studies use a tumor-informed (a priori) approach to ctDNA detection, however, do not discuss how this impacted assay detection sensitivity and data quality. Existing data suggest that ctDNA detection with a priori knowledge of tumor-specific variants is more sensitive and accurate than de novo calling (eg Chabon et al 2020 - https://doi.org/10.1038/s41586-020-2140-0).
- Consider the addition assay description columns to Table 2 to include ctDNA lower limit of detection or threshold used in each study (as described in Results), and what the specific ctDNA surrogate was (i.e., mutant KRAS allele - canonical driver mutation; custom targets - tumor informed mutations, etc.; methylation profiling) - for NGS assays, what were the targets? Custom panel for tumor-informed mutation or de novo mutation calling.
- Discussion section should include the challenge of comparing relative ctDNA fraction (MAF or VAF, as reported by Murahashi et al, etc.) between time points, patients, and studies without correcting for total cell-free DNA content (as appears to have been done by Ji et al). This latter approach is preferred as cell-free DNA levels fluctuate over time (particularly during treatment) and is typically reported as mutant haploid genome equivalent (hGE) or mutant copies per ml plasma and is derived from MAF and total cell-free DNA per ml plasma.
Specific comments:
- Line 236-237 (section titled "Outcome") - it is unclear if the four studies that did not describe the follow-up period, didn't report and clinical follow-up data, or simply didn't report when their data were collected.
- Line 263-264 (section titled "ctDNA after CRT") - Authors state that studies support previous findings but do not provide details as to how they do.
- Line 271-272 (section titled "ctDNA after surgery") - authors report increasing HR from Zhou et al associated with "ctDNA measured during CRT...preoperative ctDNA measurement HR...[etc.]" but it is unclear what the HR is based on. Please state if the HR is simply increasing with detectable vs undetectable ctDNA or if there is something specific about the ctDNA measurements that is affecting the HR (ie., increases/decrease over previous time or from baseline, etc.). The use of the terms "ctDNA measured" and "ctDNA measurement HR" is ambiguous compared to "detectable ctDNA" or relative abundance of ctDNA at the given time points.
- Line 320-321 (section titled "Discussion) - Authors state the aggregated study "...results indicate that ctDNA cannot be used pre-treatment to select patients for organ preservation or surgery." This statement is much to strong and categorical. As the authors point out themselves, the current paucity of data and lack of properly controlled studies limit our ability to make such conclusions. Consider specifying which measures are less informative based on data.0
- Line 341-345 - writing needs to be more specific. The language stating "...in nearly 50% of the trials a benefit was observed with extended follow-up..." is unclear about which trials the authors are referring to - which trials where assess in the study cited? Line 344-345 states "Loss of significant benefits was only seen in 7% of studies." again, it is unclear which studies the authors are referring to and how this work relates to ctDNA as a prognostic marker in LARC.
- Line 353-354 - The authors state that "...a surrogate endpoint would be ideal and a good candidate in rectal cancer is warranted." This sentence needs some clarity. Also, what do the authors think might be a clinically useful and cost-effective endpoint in future studies of LARC? What do the findings of the studies included in the review suggest would be a useful surrogate endpoint?
Noted typos, spelling and grammatical issues: Line 54 - additional punctuation with citations should be removed Line 62 - "PRSIMA" should be "PRISMA" Table 1, row 3, column 2 - "titel" should be "title" Line 100 - "Quality appraisal" - it is unclear if this is a new section, or a heading, etc.? Line 122-123 - Sentence is not punctuated and meaning is unclear - consider rewriting Line 168 - New paragraph? Needs indentation Line 215 - "0,15%" is inconsistent with reporting of previous decimal values using "." (ie. line 213 "0.02%") Line 327 - "litterature" should be "literature"
Reviewer 3 Report
Dizdarevic, et al., reviewed the studies of ctDNA in rectal cancer. They accessed the quality of the researches and found potential risks of bias. They found that lack of guidelines to ensure uniform reporting and studies results are diverging, thus concluding that using ctDNA as markers in diagnosis or prognosis should be cautious at this moment. I think this review raises some issues in this field that is of importance and maybe useful to researcher to guide their future studies. In general, it is a good review.
Minor issues:
- Some result descriptions should add references, for example, the “ctDNA at baseline” part, last sentence.
- Please pay attentions to the formats of the texts.
Round 2
Reviewer 2 Report
The changes made by the authors sufficiently addressed my major concerns.